# GEOMETRIC GRAPH NEURAL NETWORK BASED TRACK FINDING

## ABSTRACT

An essential component of event reconstruction in particle physics experiments is identifying the trajectory of charged particles in the detector. Traditional methods for track finding are often complex, and tailored to specific detectors and input geometries, limiting their adaptability to new detector designs and optimization processes. To overcome these limitations, we present a novel, end-to-end track finding algorithm that is detector-agnostic and can take into account multiple input geometric types. To achieve this, our approach unifies inputs from multiple sub-detectors and detector types into a single geometric algebra representation, simplifying data handling compared to traditional methods. Then, we leverage an equivariant graph neural network, GATr, to perform track finding across all data from an event simultaneously. We validate the effectiveness of our pipeline on various detector concepts with different technologies for the FCC-ee at CERN, specifically the IDEA and CLD detectors. This work generalizes track finding across diverse types of input geometric data and tracking technologies, facilitating the development of innovative detector concepts, accelerating detector development cycles, and enabling comprehensive detector optimization.

## 1 INTRODUCTION

The Future Circular Collider is one of the proposed future colliders that could follow from the Large Hadron Collider (LHC) at CERN. In the first stage, FCC-ee, an electron and a positron beam are accelerated to nearly the speed of light to collide in multiple interaction points. These collisions generate sprays of outgoing particles. Sophisticated detector systems, comprising hundreds of millions of sensors are utilized to measure information about these particles. One of the main challenges is to identify and reconstruct the 3D trajectory and kinematic properties of a charged particles propagating inside the detector's magnetic field (see Figure 1), this is referred to as tracking. Classic approaches solve this problem with combinatorial optimization methods such as Kalman filters (Ai, 2019; Bertacchi et al., 2020; Amrouche et al., 2020). However, these methods are detector dependent, and have a long development cycle, reducing their adaptability to new detector concepts. Novel approaches to tracking are needed to maximize the potential of future detectors.

Tracking is performed in two stages (Frühwirth & Strandlie, 2021), first track finding and then track fitting. Track finding is essentially an object instantiation task, which requires to identify groups of hits that form a track. Where the hits are the collection of measurements resulting from the particles interacting with the detector. Hits are usually positions, but they can be more complex geometries, such as shapes or directions. Track finding is a challenging problem: tracks can have a different geometries and varying number of hits (10-100's), a track can have missing hits in the trajectory, they can have hits in one or multiple sub-detectors, or an in-flight decay of a particle can produce an abrupt change in direction (kinked tracks) or disappear.

Track finding has been approached in multiple ways. Global methods cluster hits by transforming the coordinates to a feature space in which relevant patterns are easier to detect (Brondolin et al., 2020). Another approach is seeding and track following, usually implemented with a Combinatorial Kalman Filter (Braun & Braun, 2019). This approach evaluates combinations of hits and iteratively builds tracks (Ai, 2019; Cornelissen et al., 2008). However, it is complex due to the large combinatorics, and it requires a detailed description of the geometry and materials of the experiment. In addition, when dealing with various types of input geometric data in each sub-detector, some works

perform track finding for each sub-detector and then combine the track sections (Bertacchi et al., 2020; Cornelissen et al., 2008). For example, the Belle II experiment has two tracking detectors, and its track finding approach is composed of eight different algorithms including multiple algorithms for each tracking detector and for the combination (Bertacchi et al., 2020). Overall, these algorithms are detector dependent and have long development cycle.

Currently, there are new collider proposals with novel developments in detector technologies. For example, for the FCC-ee, there are several detector concepts under investigation (Abada A, 2019; Bacchetta et al., 2019; Bedeschi, 2021). Each different tracking component can result in multiple input geometric types, depending on the measurement. As these new detector concepts are envisioned, it is important to have adaptable tracking pipelines with fast turnaround that can cope with different sub-detector concepts, geometric input data, variations and optimizations thereof. Figure 1 shows two of the detectors which produce different track representations requiring usually different algorithms.

Novel Graph Neural Network (GNN) based methods have been recently shown to perform competitively to classic pattern recognition approaches for a generic silicon detector while remaining scalable and not depending on the geometry implementation of the detector (Ju et al., 2021; Amrouche et al., 2020). However, so far they have only been applied to silicon tracking systems, which have pointcloud-like inputs, and Ju et al. (2021) also introduce pipelines with multiple preprocessing steps.

This work presents the detector agnostic Geometric Graph Track Finding (GGTF) method. This method is a generalized geometric track finding approach that:

- Can cope with multiple sub-detectors with different input geometries from multiple tracking technologies

- Does not require the geometry and materials specifications of the detector

- Does not rely on an analytical parametrization of the trajectories.

Specifically, our end-to-end pipeline considers the hits from all tracking components and outputs a set of tracks. A critical new component is a geometric algebra representation of the data that which allows multiple geometric types and a GNN, GATr (Brehmer et al., 2024), that exploits the symmetries of the detector through equivariance. The performance of the algorithm is assessed using two of the FCC-ee baseline detector concepts, the IDEA detector (Abada A, 2019; Bedeschi, 2021) and the CLD detector (Bacchetta et al., 2019).

## 2 RELATED WORK

**Legacy tracking approaches**   Two main approaches are employed in track finding algorithms: coordinate transformation (global methods), and seeding and track following approaches. The simpler example of the first set, coordinate transformation, is conformal tracking. Conformal tracking (Hansroul et al., 1988) is a circle fitting method that transforms circles in the plane passing through the origin into straight lines. More complex transformations are the Hough transform (Duda & Hart, 1972), which finds clusters of points that lie close to a parametric curve reducing tracking to finding intersection points, and the Retina algorithm (Ristori, 2000). However this algorithms tend not to reconstruct more complex tracks including: tracks displaced from the vertex or kinked tracks. For this reason, seeding and track following is the more common approach to track finding. Conformal tracking can be improved using this type of approach. Brondolin et al. (2020) use the cellular automaton (CA) algorithm to take these deviations from the circular path, such as those due to multiple scattering or displaced vertices into account. The CA algorithm uses a seeding procedure followed by cell extrapolation. In other approaches, the seeding algorithm iteratively finds triplets of hits that are likely to belong to the same track. The track candidates are then built from the seeds using a Combinatorial Kalman Filter (Braun & Braun, 2019). These types of approaches give high accuracy but are computationally demanding. Some of these algorithms have complex implementations as they use the geometry to calculate deviations from the expected path due to material interactions. This requires the algorithm to interact with the geometry of the detector for every step which can be costly (Ai et al., 2022; Ai, 2019).

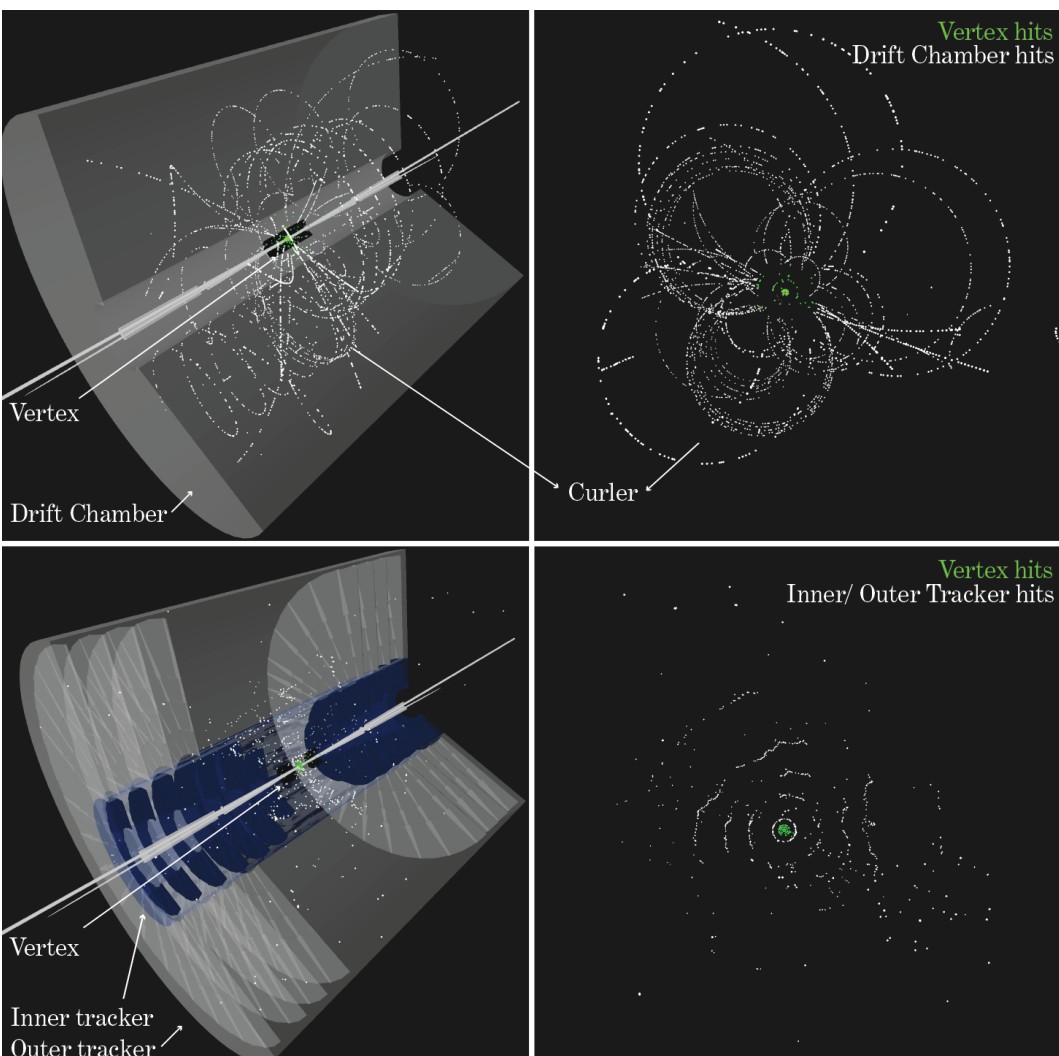

Figure 1: Top left: Simulated collision event showing the hits (in white) in the tracker of the IDEA detector. In the drift chamber volume each hit is composed of two position coordinates due to the ambiguity in the measurement (more details are given in Section 4). Top right: front view of reconstructed hits. The circular shrinking pattern on the left corresponds to a low energy particle curling inside the detector while losing energy. The straight tracks correspond to higher energy particles. Bottom left: reconstructed hits in the CLD detector of the same event, the vertex-detector is shown in black, the inner tracker layers are shown in blue and the outer tracker disks are shown in grey. Bottom right: front view of the hits in the CLD detector.

An extensive review of classic pattern recognition algorithms is out of the scope of this paper, we refer the reader to (Frühwirth & Strandlie, 2021) for an in depth review. However, it is worth noting that track finding in real experiments, such as the Belle II (Bertacchi et al., 2020) applies a mix of these algorithms to obtain the best tracking performance.

**ML-based tracking**    Track finding can be posed as an object detection problem. Multiple approaches exist for object detection for point-cloud datasets, some focus on regression of bounding boxes Yang et al. (2019); Hou et al. (2019); Wang & Solomon (2021), others find instance queries using a voxelized representations of the point-cloud Kolodiazhnyi et al. (2024). However, these approaches are not directly applicable to track finding as the bounding boxes are hard to define due to the geometry of the tracks, and the voxel representation would merge multiple tracks. For these

reasons, the ML-based tracking approaches have focused on the tasks of edge prediction and discovery of connected components as a second step Denby (1988); Stimpfl-Abele & Garrido (1991); Ju et al. (2021); Farrell et al. (2018). These methods suffer from the costly post-processing step of finding connected components in a graph. Earlier works also considered Recurrent Neural Networks to perform a binary hit classification graph model that works on a subset of the event to extract tracks iteratively. More recently, Lieret & DeZoort (2024) use a GNN based approach with a contrastive learning loss to create a separable embedding space where they can perform clustering. This method improves the performance over the connected components baseline, and is similar to some object detection approaches based on similarity metrics Wang et al. (2018; 2019).

GNN based approaches (Farrell et al., 2018; Ju et al., 2021; Lieret & DeZoort, 2024) have focused on the dataset proposed in (Amrouche et al., 2020) which considers pixel and strips tracking components. One of our key contribution is to generalize the method of track finding using GNNs to more complex tracking detectors.

## 3 PRELIMINARIES

**Problem statement**   An event is a collection of measurements in a detector. The detector registers the signals deposited by the interaction of particles with its sensitive components, referred to as hits. The role of a tracking algorithm is to identify sets of hits produced by single particles and to derive information on the particles trajectory based on those. For a set of hits in a given event, $\mathcal{X}$, composed of hits from different tracking components $\mathcal{X} = \{X_v, X_i, X_o, ...\}$, (vertex, inner tracker, outer tracker, endcap inner tracker,...) we want to obtain a set of tracks that approximates the target set of charged particles. The geometry of the detector is cylindrical, with a magnetic field along the $z$ direction. We define the $(x, y)$ as the transverse plane , and the $z$ as the beam direction. Figure 1 shows an example set of hits for different detectors. In this work, this is framed as an object instantiation problem.

**Trajectories**   In a detector with a homogeneous magnetic field, the trajectory of a particle forms a helix, with varying curvature depending on its momentum. Particles with higher momentum result in an arc, while lower momentum particles can curl inside the detector. As they lose energy, the curvature of the helix increases, generating the shrinking displaced circles observed in Figure 1 top.

**Representations**   The inputs from the different tracking detectors, as well as the geometries of the tracks, are detector dependent. For a silicon detector, each hit $h = (\boldsymbol{x}, s)$ consist of a set of 3D coordinates $\boldsymbol{x} \in \mathbb{R}^3$ and a scalar $s$ corresponding to the subdetector index. For other detector technologies, other information might be available. For example in a drift chamber, a gaseous detector filled with wires which register ionization signals, each hit leads to two sets of coordinates, $(\boldsymbol{x}_{\text{left}}, \boldsymbol{x}_{\text{right}})$, due to inherent left-right ambiguity in the measurement. This richer data structure, which steps away from point-like track representations, make the application of classical approaches complex.

In addition, tracks might have very different geometries in different detectors. For a silicon detector like CLD a particle will leave around 10-15 hits per track (Bacchetta et al., 2019), while for a drift chamber each particle can generate O(100) hits. Not only the number of hits are different, but also the presented geometries. A drift chamber like detector allows to capture particles of very low energy looping through the detector due to the magnetic field, as can be seen in Figure 1 left. Lower momentum particles have smaller curvature radius leading to spiral trajectories, which we will refer to as curlers. These curlers represent a complex reconstruction tasks for classical algorithms, which usually focus on the high momentum regime (higher curvature radius tracks).

## 4 GEOMETRIC-BASED GENERALIZED TRACK FINDING

Our main contribution is the GGTF pipeline, which transitions from the hits of the different tracking detectors to a set of tracks. Our pipeline uses a novel geometric GNN, the Geometric Algebra Transformer (GATr), (Brehmer et al., 2024) which represents the inputs in a projective geometric algebra respecting equivariance, and the object condensation loss (Kieseler, 2020) to cluster hits belonging to the same object close in an embedding space, as in (Ju et al., 2021). Unlike (Ju et al.,

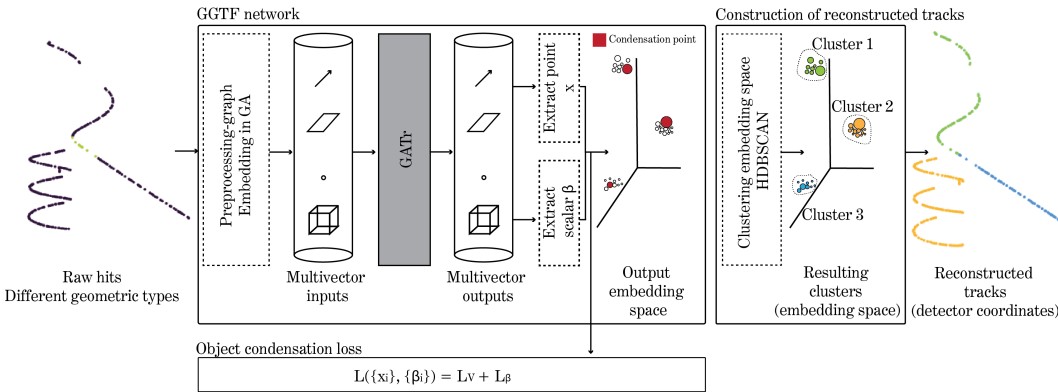

Figure 2: Overview of the GGTF end-to-end approach for track finding. The raw hits are represented as multivectors in the geometric algebra. The multivectors are transformed using the GATr. The output of the network is a coordinate in 3d space and a scalar, $\beta$, obtained from the output multivector by extracting the point and scalar components. Each ground truth track (object) should result in a high $\beta$ node or condensation point in the embedding space, pictured filled in red. The outputs of the network are used by the object condensation loss to determine attractive and repulsive potentials between nodes to further allow a separable output embedding space as described in Section 4. Finally a clustering step determines the track candidates by forming subsets of hits.

2021; Lieret & DeZoort, 2024), our architecture does not require edge filtering which reduces the memory constraints. Also, our pipeline allows for more types of geometric inputs, unlocking this new approach for more tracking detectors such as the drift chamber.

**Representation of hits in the graph**   As an initial step, the set of hits for each event is converted into a graph. However, unlike traditional approaches, the GATr architecture allows to represent the raw hits as different geometric objects and transformations in the projective geometric algebra $\mathcal{G}_{3,0,1}$. The elements of the algebra are 16-dimensional multivectors, these have direction, sign and can be linearly combined (Brehmer et al., 2024). This method allows to have as an input planes, lines, point or scalars. For example, we choose to represent the hits from the vertex detector as points (trivectors) and the hits from the drift chamber as vectors. This vector goes from the left to the right hit coordinates, therefore accounting for the ambiguity in the measurement.

**Geometric Algebra Transformer**   The hits from the different systems are then unified in the multivector format, which represents the different geometrical objects in 16 dimensions. Then, we use the GATr architecture as shown in Figure 2. It consist of multiple transformer blocks. Each block has a *LayerNorm*, an *equivariant multivector self-attention* layer, a second *LayerNorm*, an *equivariant multivector MLP* with *geometric bilinear interactions*, and another residual connection. This architecture is $E(3)$ equivariant. The $E(3)$-equivariant linear layers map multivectors using grade projections and multiplation with a homogenous basis vector. The geometric bilinears allow to build more expressive networks by allowing grade mixing. The network outputs multivectors which are used by the object condensation loss, defined below.

**Loss function**   Modern approaches to reconstruct multiple objects focus on predicting a bounding box per object as an output (Wang & Solomon, 2021; Girshick, 2015). However, due to the helical trajectory of the tracks it would be hard to parameterize these boxes. On the other hand, the object condensation approach for multi-object detection (Kieseler, 2020) allows to reconstruct an unknown number of objects through a dedicated loss function without the use of bounding boxes. The intuition behind this approach is as follows: the loss function requires the output of hits belonging to the same track to be close in a low dimensional embedding space and those belonging to different tracks to be far away. In this way, in the embedding space the tracks are easily separable.

The object condensation loss, $L$, is composed of two terms: a potential loss term $L_V$ to drive the position of the coordinates in the embedding space, and $L_\beta$ which is used to avoid a minimum of

$L_V$ and will be described below. First, in order to define these losses we need to define the attractive and repulsive potentials. The force affecting every node, $j$, belonging to an object $k$ is

$$q_j \nabla V_k(x_j) = q_j \nabla \sum_{i=1}^{N} M_{ik} V_{ik}(x_i, q_i) \tag{1}$$

$q_j$ is a charge per node and $x_j$ are the coordinates of the node on the output embedding space. The mask $M_{ik}$ is 1 if $i$ belongs to $k$ and 0 otherwise. Therefore, the network is trained to predict a scalar $0 < \beta_i < 1$ and a coordinate $x$ per node. The scalar is used to define the charge $q_j$ with an increasing gradient:

$$q_i = \arctan^2 \beta_i + q_{min}, \tag{2}$$

$q_{min}$ represents the minimum charge and avoids the minimum $q_i = 0$. In order to avoid calculating $N$ matrices to calculate the potential, the potential affecting nodes belonging to an object is approximated by the potential of the node with the highest charge in the object, $\alpha$. For each track/object $k$, the attractive loss is defined as:

$$L_{\text{attractive},k} = \frac{1}{N_k} \sum_{j=1}^{N} q_j (M_{jk} \check{V}_k(x_j)), \tag{3}$$

where $N_k$ is the number of hits that belong the object. We define the repulsive loss per object as the sum of the repulsive potentials, $\hat{V}$, of all hits that do not belong to the object:

$$L_{\text{repulsive},k} = \sum_{j=1}^{N} q_j (1 - M_{jk}) \hat{V}_k(x_j), \tag{4}$$

unlike the attractive loss, it is not divided by the total number of contributing hits as this results in more separable embedding spaces. We refer to Appendix A.1.1 for a comparison of different loss definitions, such as the hit-based definition in Kieseler (2020). Therefore the loss corresponding to all tracks can be written as follows

$$L_V = \sum_{k=1}^{K} (L_{\text{attractive},k} + L_{\text{repulsive},k}). \tag{5}$$

Then, $L_\beta$ is defined to enforce one condensation point per track (left side):

$$L_\beta = \frac{1}{K} \sum_{k} (1 - \beta_{\alpha k}) + \frac{1}{N_B} \sum_{i}^{N} n_i \beta_i \tag{6}$$

it also minimizes the $\beta_i$ component for noise hits ($n_i = 1$), where $N_B$ are the number of hits that are not assigned to a reconstructable particle. The full loss function is $L = L_V + L_\beta$.

Starting from the output multivectors of the GATr network, we define the coordinates of each node as the point component extracted from the multivector and the $\beta$ for each node as the scalar component. This is represented on the right side of Figure 2.

**Final track reconstruction**   During inference, the extracted point component of each multivector, the output embedding is clustered using the HDBSCAN clustering algorithm (McInnes et al., 2017), as this algorithm is able to reconstructs clusters of different densities. We refer to Section 5.1 for a comparison of different clustering algorithms. Finally, the hits inside each output cluster define a reconstructed track as shown in Figure 2 right.

## 5   EXPERIMENTS

**Metrics**   In order to measure the performance of the various algorithms, each reconstructed track is matched to a ground truth particle to which it shares the largest number of hits.

- *The track hit purity* is the ratio of the number of hits in the track that belong to the MC particle and the total number of hits of the reconstructed track.

- The *track hit efficiency* is the ratio of the number of hits in the track that belong to the MC particle and the total number of hits this particle left in the detector.

The *tracking efficiency* is the probability to reconstruct a track. It can be defined in multiple ways, we will consider the following variants:

1. The percentage of reconstructable charged particles that lead to a reconstructed track with track hit purity greater than 75%, as in Bacchetta et al. (2019) .

2. The percentage of reconstructable charged particles with both ratios, the track hit purity and the track hit efficiency, above 50%, as defined in the accuracy tracking challenge Amrouche et al. (2020). Tracks are further separated in four categories. *Good*: both ratios above 50%, reconstructed in this definition. *Split*: track hit efficiency below 50%, track hit purity above 50%, so only a fraction of the track is reconstructed. *Multiple*: track hit efficiency above 50%, but track hit purity below 50%, typically due to the aggregation of multiple particles. *Bad*: both below 50%.

3. For the CLD detector as there are only 10-15 hits per track, we consider the percentage of reconstructable charged particles that match with a reconstructed track with at least four hits (the minimum number of hits for track fitting, the second tracking step).

We evaluate the performance of our algorithm by showing the efficiency vs particle proximity and track displacement. *Particle proximity* is the smallest angular distance, $\Delta_{MC}$, between the Monte Carlo particle associated to the track and any other MC particle. The angular distance between two particles can be defined as a function of the azimuthal angle $\phi$ and the pseudorapidity $\eta$ as $\Delta_{MC} = \sqrt{\Delta\phi^2 + \Delta\eta^2}$. Reconstruction of *displaced tracks* is critical for the identification of longed lived particles. These tracks are originated from a secondary vertex rather than from the primary interaction point e.g b and d mesons which can travel a measurable distance before decaying. The displacement of a track can be measured looking at the production vertex radius, i.e. $R = \sqrt{x^2 + y^2}$.

## 5.1 CLD DETECTOR

We start the demonstration of the GGTF pipeline and performance with the CLD detector. This detector is an all-silicon tracker, therefore an easier case scenario as all hits are of the same type, and can be compared to a heuristic based algorithm: the conformal tracking (Brondolin et al., 2020; Bacchetta et al., 2019). The input are the hits left by the particles in the different subdetectors: vertex, inner tracker and outer tracker. We generate a training dataset with 320k events resulting from a $e^+e^- \to Z \to q\bar{q}$, $q = u, d$ decay, at a center of mass energy of 91.2 GeV/c; a validation dataset with 10k events; and a regular evaluation set with 10k events for each detector concept. The experiment, dataset details, and different cuts applied to be comparable to the baseline are described in detail in Appendix A.1.1.

In the top left and center Figure 3 we show the track hit purity and track hit efficiency respectively. Conformal tracking has a higher track hit purity but captures up to 40% less hits than GGTF in the low $p_T$ regime. This shows in the different efficiency definitions. Using the definition based only on purity, definition 1, the tracking efficiency is slightly lower than the baseline for low $p_T$, shown in bottom left of Figure 3. If we consider the second definition, based on track hit purity and efficiency, GGTF has an overall higher tracking efficiency, shown in bottom center of Figure 3. The discontinuity observed at 700 MeV can be explained by the fact that this is the minimal transverse momentum needed to reach the outside of the tracker, therefore this threshold set the limit for curlers.

We use the tracking efficiency definition 3 to study the performance on displaced tracks and particle proximity. This definition allows to compare both algorithms on equal grounds as it uses a loser definition of reconstructed track. GGTF outperforms the baseline for displaced tracks reconstruction, as shown in the top right of Figure 3. This is due to the conformal tracking algorithm assuming that all tracks originate from a common point. Therefore, these displaced tracks do not align with straight lines in the conformal plane, which leads to failure of the conformal tracking algorithm. The percentage of fake tracks is very similar for both algorithms: 4.8% for GGTF and 5.2% for conformal tracking. Since the track fitting stage is not performed, the distribution per $p_T$ is not available.

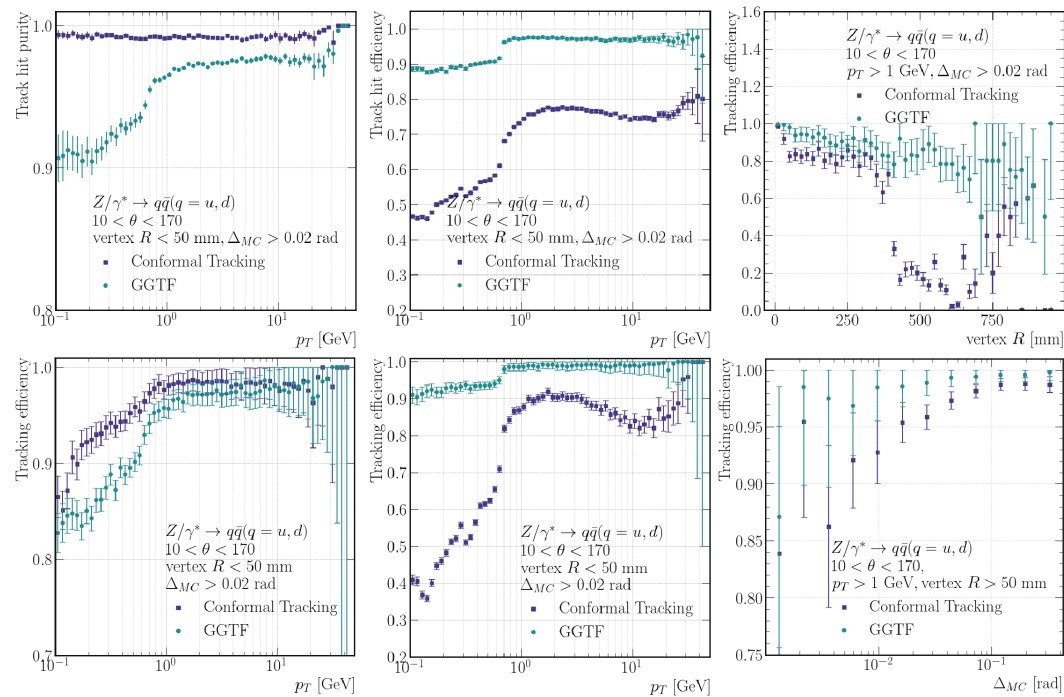

Figure 3: Results for the CLD detector. Top left: track hit purity as a function of $p_T$. Top center: track hit efficiency as a function of $p_T$. Top right: tracking efficiency as a function of the production vertex radius (def. 3). Bottom left: tracking efficiency (def. 1) as a function of $p_T$. Bottom center: tracking efficiency (def. 2) as a function of $p_T$. Bottom right: tracking efficiency as a function of particle proximity.

Finally, the bottom right of Figure 3 shows the efficiency with the most inclusive definition 3 as a function of particle proximity. Overall, this follows the expected trend, if the closest track is further, it is easier to detect the tracks and GGTF improves the performance for really close tracks over the baseline.

We compare the performance of different clustering algorithms (HDBSCAN (McInnes et al., 2017), DBSCAN (Ester et al., 1996), beta-based object condensation (OC) (Kieseler, 2020)) in Figure 6. HDBSCAN has similar performance to OC but is chosen for our pipeline due to its lower execution time. DBSCAN has lower performance, as expected, as it does not consider clusters of varying densities.

Figure 5 shows the classifications of tracks for reconstructable particles as described in Section 5. The conformal tracking approach presents more split tracks while the GGTF can reconstruct a higher percentage of 'full' tracks.

## 5.2 IDEA DETECTOR

Next, we turn towards a more complex detector involving multiple tracking technologies and producing more complex geometries. We study the IDEA detector which has a silicon inner vertex detector and a drift chamber. The drift chamber is filled with gas and wires that collect the ionization signal of particles. Each particle can leave O(100) hits in the detector, which is an order of magnitude larger than for CLD. The IDEA detector does not have a baseline algorithm available. We therefore use these results to illustrate the algorithm versatility in terms of detector technology. The dataset decay and size are the same as those described in Section 5.1 for the CLD detector. See Appendix A.1.2 for all the experiment details.

The results are shown in Figure 4. Figure 4 shows that GGTF provides good tracking efficiency (definition 2). The curling limit of $p_T$ is shown with a blue vertical line. As expected, the tracking

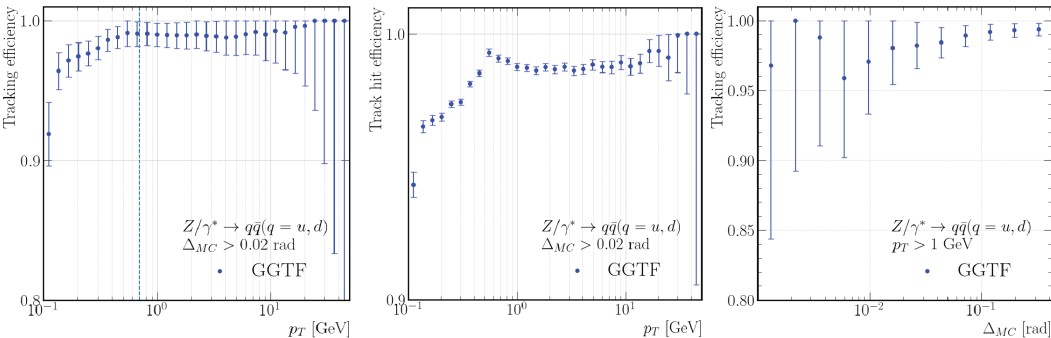

Figure 4: Results for the IDEA detector. Left: tracking efficiency (def. 2) as a function of $p_T$. Center: percentage of hits of the MC particle captured by the reconstructed track as a function of $p_T$. Right: tracking efficiency as a function of particle proximity.

efficiency is lower for lower $p_T$ particles as these are curling in the detector leaving very long tracks which are hard to recover fully. This is shown in the center of Figure 4. These results are comparable to the tracking efficiency shown for the Belle II detector Bertacchi et al. (2020).

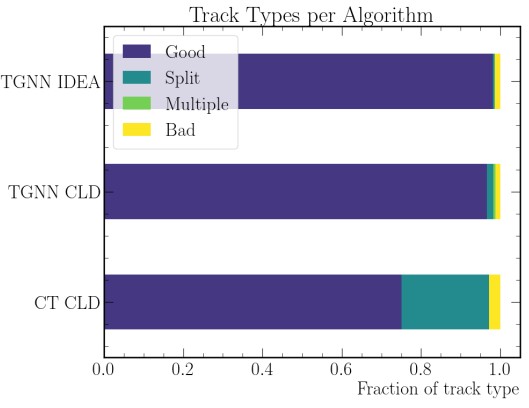

Figure 5: Distribution of the reconstructed tracks with different definitions as described in section 5 for the GGTF and the baseline approach

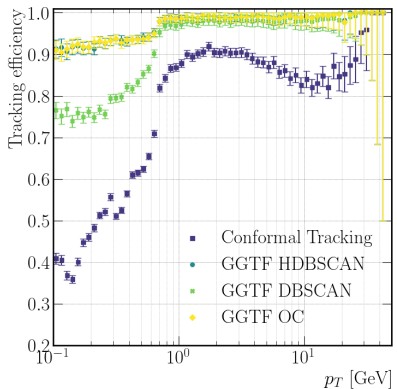

Figure 6: Analysis of the GGTF using different clustering schemes.

## 6 CONCLUSIONS

GGTF is a high-efficiency general track finding method for multiple sub-detectors with different input geometries. It is able to find varied track geometries via an equivariant GNN. The success of GGTF indicates that highly parameterized algorithms are likely unnecessary and can be replaced with suitable GNNs. This new pipeline significantly simplifies the application of track finding to optimization studies as the development cycle is largely reduced. An important future step is to evaluate the algorithm impact on the track fitting stage, as its higher track hit efficiency could improve the accuracy of the track parameters. Beyond the current applications, we suggest multiple research directions to address the current limitations. For example, the algorithm was evaluated without background, adding background would result in several 'noise' hits, thereby degrading the track hit purity and efficiency and increasing the probability of reconstructing fake tracks. Finally, scalability of the algorithm needs to be improved to apply this method to full pile-up events, where the number of hits can be up to an order of magnitude larger.

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

# A APPENDIX

## A.1 EXPERIMENTS

**Dataset** Raw hits are obtained by performing simulation using the Key4hep (Brondolin et al., 2024) framework, a software ecosystem for future colliders (Brondolin et al., 2024). First, the simulation of the collisions, production and decay of the Z boson are generated with PYTHIA8 (Sjöstrand et al., 2006). The considered decay is $e^+e^- \rightarrow Z \rightarrow q\bar{q}$, $q = u, d$, at a center of mass energy of 91.2 GeV/c. PYTHIA8 also evolves the produced particles, performing parton showers and hadronization to produce a final set of outgoing particles. These particles are then propagated through the detector using the DD4hep (Frank et al., 2014) toolkit which provides a convenient interface to Geant4 (Agostinelli et al., 2003). Our approach is applied to the two detector concepts under consideration. We generate one dataset for each detector concept and describe their differences below.

We generate a training dataset with 320k events; a validation dataset with 10k events; and a regular evaluation set with 10k events for each detector concept.

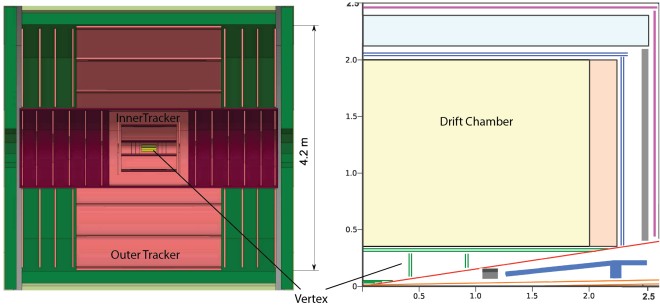

Figure 7: Left: layout of the CLD tracking system from Bacchetta et al. (2019). Right: layout of the IDEA tracking system from Gaudio (2023).

## A.1.1 CLD DETECTOR

The CLD detector (Bacchetta et al., 2019) has an all-silicon tracker. The tracking system consists of a vertex pixel detector (V), an inner tracker with three barrel layers (ITB) and seven forward disks or endcap (ITE), and an outer tracker with additional three barrel layers (OTB) and four discs (OTE). We use version *CLDo2v05* of the CLD detector geometry (Gaede et al., 2024).

**Input format** The dataset includes hits from each tracking sub-detector, reconstructed tracks from the conformal tracking algorithm and association of each hit to a Monte Carlo particle to build the set of target tracks to train the model on. Each input hit has global detector coordinates $(x, y, z)$ and a label of the subdetector they hit $\{V, ITB, ITE, OTB, OTE\}$. Additionally, hits created by MC particles with less than 3 hits in total are labelled as noise hits, and the algorithm does not try to reconstruct these tracks.

**Models** For the GATr we consider the architecture in Brehmer et al. (2024). We embed the hits of all sub-detectors as trivectors. We use 10 attention blocks, 16 multivector and 64 scalar chanels, and 8 attention heads, resulting in 1.1 million parameters. The $\beta$ is extracted as the scalar component of the multivector, and the coordinate in the embedding space as the trivector component.

**Training** All models are trained by minimizing the object condensation loss. We train for 20 epochs with the Adam optimizer, using a batch size of 8, and a step scheduler for the learning rate from $10^{-3}$ to $10^{-6}$.

**Baselines** We compare our approach to state-of-the art track finding algorithms. The reference track finding approach for this detector which is based on conformal tracking and cellular automaton

(Brondolin et al., 2020; Bacchetta et al., 2019). In order to reproduce the results of Bacchetta et al. (2019), we apply the following cuts to the reconstructable particles. For the efficiency calculations the cuts are: $10° < \theta < 170°$, production radius smaller than 50 mm and a minimum particle proximity larger than 0.02 rad. For the production vertex radius the applied cuts are: $p_T > 1$ GeV, $10° < \theta < 170°$, a particle proximity larger than 0.02 rad. For the particle proximity the applied cuts are: $p_T > 1$ GeV, $10° < \theta < 170°$, and a production vertex radius, $R < 50mm$.

**Additional plots** Percentage of unique hits: A given particle can leave more than one hit per layer, in which case, only one of the hits is considered unique. Evaluating the percentage of unique hits allows to determine which algorithm captures more relevant hits. In addition, capturing the maximum amount of unique hit improves the second stage of tracking, track fitting. Figure 8 shows that GGTF captures up to 40% more unique hits for lower $p_T$ particles and 20% for higher $p_T$.

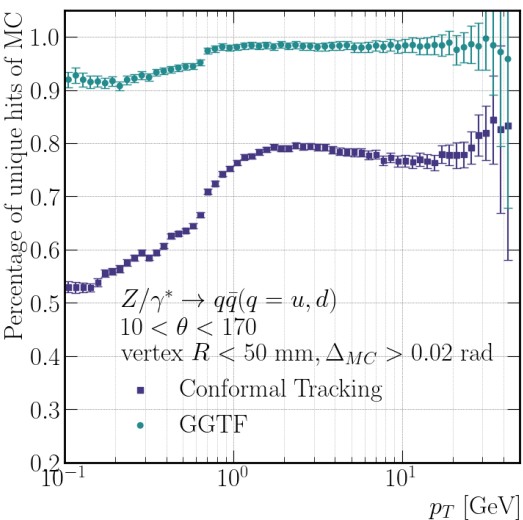

Figure 8: Percentage of unique hits vs $p_T$ of the particle.

**Comparison to other GNN architectures** We compare the GATr architecture to the Gravnet architecture Kieseler (2020) and the architecture proposed in Lieret & DeZoort (2024). The approach presented in Lieret & DeZoort (2024)consist of first constructing the graph based on geometric constraints, then filtering the edges using a fully connected neural network, and finally using object condensation with a GNN. The last GNN takes advantage of the weights learned by the edge filtering. The geometric edge based constraints implemented in their paper are very biased towards high $p_T$ tracks, as these are the 'tracks of interest' for their application. We observe that due to the helical trajectories of low $p_T$ tracks the edges are completely removed, resulting in many disconnected components if the geometric edge based constraints are applied. With our reproduced implementation we are not able to obtain convergence during the training for our dataset and this results in clustering spaces that can not be clustered. The Gravnet architecture is substituted in our pipeline, and the loss function and the clustering step remained unchanged. The results comparing the Gravnet architecture to the GATr are shown in Figure 9. Gravnet shows degraded performance compared to GATr. We attribute this to the lack of geometric representation in the network and the complexity of the geometry of the tracks.

**Comparison of different loss functions** We compare the performance of the GGTF pipeline under different loss definitions. We consider the object condensation loss per hit in the event, per track/object in the event and the loss defined in Section 4. Since different tracks might have a very different number of hits the per track definition can help balance the training for tracks with a larger number of hits. For the condensation loss per hit, the $L_V$ term is different to the one presented in

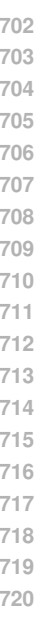

Figure 9: Comparison of the track efficiency (definition 2) vs $p_T$ including the Gravnet model.

Eqn. 5 and the $L_\beta$ term is as defined in Eqn. 6. The potential term is defined as:

$$L_V = \frac{1}{N} \sum_{j=1}^{N} q_j \sum_{k=1}^{K} (M_{jk} \check{V}_k(x_j) + (1 - M_{jk}) \hat{V}_k(x_j)). \tag{7}$$

Therefore each hit/node contributes equally to the loss function. For the condensation loss per object, the attractive potential is the same as the one defined in Eqn. 3, and the repulsive potential is defined as:

$$L_{\text{repulsive},k} = \frac{1}{\hat{N}_k} \sum_{j=1}^{N} q_j (1 - M_{jk}) \hat{V}_k(x_j), \tag{8}$$

where $\hat{N}_k$ is the number of nodes in the event that do not belong to object k. The $L_\beta$ term also remains unchanged for this loss definition. The results of training our pipeline with the different loss functions is presented in Figure 10. The best performing loss is the loss chosen for the main pipeline, described in Section 4. This loss uses a per shower attractive potential and a repulsive potential to all points not belonging to the shower without normalization. This points to the repulsive potential definition having a larger impact on the performance, as the per track loss which shows the worst performance, shares the attractive potential with the GGTF loss function.

### A.1.2 IDEA DETECTOR

The tracking system of the IDEA detector (Abada A, 2019; Bedeschi, 2021) is composed of an inner vertex detector, a drift chamber as main tracker and a silicon wrapper providing an additional precise measurement at large radius. The modeling of the silicon wrapper geometry being not available at the time of this study, only the vertex detector and drift chamber have been included in the simulation. The vertex detector is a silicon pixel detector surrounding the beam pipe for the precise determination of the impact parameter of charged tracks. The hits resulting from this detector include global coordinate information as in CLD. The drift chamber is a full-stereo, unique volume, highly granular cylindrical chamber. This tracking detector allows to measure the $d_{\text{along}}$ along the wire and the distance to the wire, $d_w$. Therefore each hit can be described as $h = (w, d_w, d_{\text{along}})$, where $w$ is the index of the wire. Using this information and the wire's information this can be converted to global coordinates. However, like in planar drift chamber there is an inherent left-right ambiguity of the spatial position relative to the sense wire, so that each hit has a mirror hit (Frühwirth & Strandlie, 2021). Therefore each hit corresponds to two sets of coordinates, $(x_{\text{left}}, x_{\text{right}})$, and the ambiguity can only be resolved after the track finding step (Frühwirth & Strandlie, 2021). Our pipeline allows to account for this ambiguity (sec: 4). The hits of the IDEA detector are stored as:

$$h_i = (x, y, z, 0) \tag{9}$$

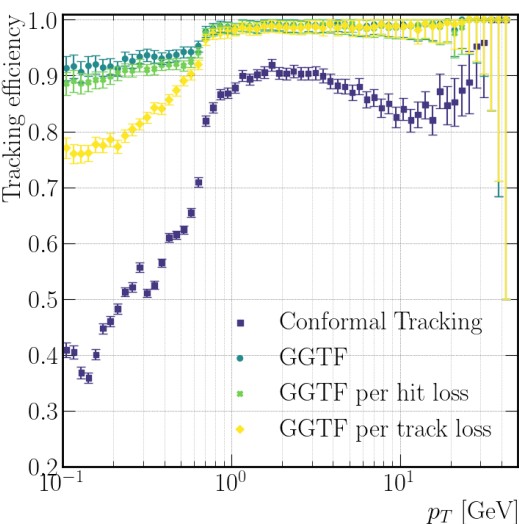

Figure 10: Comparison of the evaluation of track efficiency (definition 2) vs $p_T$ for different loss functions.

if they belong to the vertex. And,

$$h_i = (x_{\text{left}}, y_{\text{left}}, z_{\text{left}}, x_{\text{right}}, y_{\text{right}}, z_{\text{right}}, 1) \qquad (10)$$

if they belong to the drift chamber.

**Models** The model architecture has the same parameters as those described in Appendix A.1.1.

**Training** All models are trained by minimizing the object condensation loss. We train for 10 epochs with the Adam optimizer, using a batch size of 8 and a step scheduler for the learning rate from $10^{-3}$ to $10^{-6}$.

**Baselines** The IDEA detector does not have a baseline algorithm available. The Belle II experiment also has a tracking system composed of a vertex detector and a drift chamber. The results are not directly comparable as the detectors have different geometries and the set up of the wires of the drift chamber is different (Bertacchi et al., 2020). However we use the results presented in Bertacchi et al. (2020) as a guideline.

