# OpenReview forum: "Geometric Graph Neural Network based track finding"
_ICLR.cc/2025/Conference — Submitted to ICLR 2025_

### Official Review · Reviewer_oxLB · 2024-11-01

**Soundness:** 2
**Presentation:** 3
**Contribution:** 2
**Rating:** 5
**Confidence:** 3

**Summary:**

This paper proposes to use  machine learning approach to perform track finding in particle physics.Itdesigned a detector-agnostic model that can beadaptedto differentdetector types.

Ageometric algebra representation in which hitscanbe represented as different geometric objects and transformationsis used to handle input from differentdetectorsandan equivariant graph neural network(GNN)calledGATr, capable of handling multiple geometricinput types from various sub-detectorsis utilized toidentify track pattterns.

The key components are the  following:
1. The unifiedGeometric Algebra Representationsimplifiesintegrating diverse detector data, providingconvenience for subsequent procedures.

2. Equivariant GNN (GATr)utilizes geometric propertiesofdetector data, helping identify patterns regardless ofdetector design. The Object Condensation Lossclusters hits belonging tothe same object close in the embedding space,allowing more efficient reconstruction.

3. The experiments demonstrate the universality of theproposed model and the better track findingresultbroughtby thewell-designed model architecture.

**Strengths:**

The unified method for track findingcan largely facilitate track finding in particle physicsThe paper is well written, easy to read and self-contained despite building on a larger bodyof particle physics.

The basic idea appearsquiteusefulin particle physics: The ability to performdetector-agnostic track finding brings about applicability across different experimental setups. The unified geometric representation simplifies data handling and reduces complexity compared to traditional methods. The experiments demonstrated high tracking efficiency and accuracy on both the CLD and IDEA detectors, showcasing its versatility and robustness inreal-world applications.

**Weaknesses:**

1.It seems that the main components of the proposedmodelare borrowed form existing works(i.e. Geometric algebra transformer)and the modules proposed by authors aresomewhatlimited. Is it a combination of existing workfor a particle physics problem?

2.Lackof explanations of model details and technicalroute selection. More extensiveexperiments for ablative study of key components areneededto demonstrate the contributionofeach part.

3.The key aspects are notvery relevant withmachinelearningas anICLR submission. Moreanalysis from the machine learning perspective is needed. I think thesubmitted main contentorrepresentation does fit for a machine learning conference well.

**Questions:**

Canyou provide some related papersinmachine learning conferencesfocusing on trackfinding in particle physicsorsimilartopics?

Arethere any ablativeexperimentsdemonstratingthe contributionsofthemajor parts tothe final performance?The experiments just shows improved tracking trackingefficiencyunder different definitions andwhichpart takes that increment are notwell-addressed.

L270~L272: output of hits belonging to the same track to be close ina low dimensional embedding space and thosebelonging to different tracks to be faraway.
Have you tried some more advancedcontrastive learning methods toachievethis?

---

> ### Author Response · Authors · 2024-11-20
>
> Thank you for your comments. Following the above order:
> 1. Our work builds on the existing tools for the very relevant problem of track finding, and creates a new SOTA baseline. However, it also shows that a geometric-GNN together with the object condensation loss can be used to perform object detection for complex geometries.  Other approaches to object detection are:
> a) Regression of bounding boxes (BoNet, 3D-SIS, Object DGCNN). However, these are not fit for our problem due to the varied geometries of tracks, and the complexity of correctly defining bounding boxes for different detectors.
> b) Feature extraction followed by pooling of voxels, and query selection (OneFormer3D). The different geometries of tracks and their possible closeness in space does not allow for good separation in voxels or superpoints, unlike Lidar like datasets.
>
> Therefore, the particularity of our data make this problem relevant for object detection challenges for the ML community.
> Additionally, to address the first question, a ML challenge was organized in 2018 to explore new solutions to track finding “Tracking Machine Learning Challenge” and the results were presented in Neurips 2018. However, this is still a small, unexplored field where the first contributions are just appearing, unlike other approaches typically applied to molecular graphs (ZINC), proteins or products (OGB).
> For reasons, we believe this work is a good candidate for the “applications to physical sciences” track of ICLR.
>
> 2. To address the lack of explanations to the model selection route, we introduced the following changes:
> a)  We address approaches such as bounding boxes or query selection in the related work section
> b) We realize an ablation study of different parts of our pipeline to characterize the performance of each part:
> i) The loss function: We consider different variations of the loss function and show their impact in Appendix A.1.1 (Figure 10)
> ii) The clustering step: we consider three different clustering methods, HDBSCAN, DBSCAN and beta-based clustering (as in the object condensation paper), and report the results of tracking efficiency vs pt of the different methods in Figure 7. We develop the findings in the results section.
> iii) GNN Architecture: we consider different GNN architectures to compare against GATr. We compare against Gravnet and Edge classification+a GNN based on interaction network layers, as proposed in  Lieret & DeZoort, 2024. Both GNNs  are trained with the same loss function as GATr and evaluated with the HDBSCAN clustering. The performance of Gravnet is compared to that of GATr and the heuristic baseline, the results are shown in the appendix. The performance is largely degraded as the training of the network does not result in good output embedding spaces. The results are even more extreme with the second approach (EC+interaction layers GNN), where the training does not converge and the clustering does not find relevant tracks. Therefore we can not produce a track efficiency for this architecture.  It is not clear why the performance in this case is degraded, as Lieret & DeZoort show good performance for high energy tracks, and we will continue to investigate this architecture.
> c) Regarding the question on the loss function, the object condensation loss is  one of the recent approaches (2020) on multi-object detection. It is a more complex loss than just calculating the distances being ‘close or far away in an embedding space’, and this was added as an introduction to the loss to provide a high level understanding. The loss section 4.”loss” has been developed in more detail in this iteration. Fig 1 (from the object condensation paper)  shows the effective potential of a node belonging to a condensation point in the center while it’s being repelled by the other condensation points. A simpler version of this loss is described in ASIS (2019) or SGPN (2017), where the condensation point in the object condensation loss (the point with the highest beta) is simplified to be the mean of the class instance. To the best of our knowledge, object condensation is one of the most advances approaches for contrastive loss for object detection, but if you are aware of others please let us know and we will work to implement it and compare against it.
>
> 3. Although ICLR is an Machine Learning  conference, we submitted to the track of applications to physical sciences, therefore we believe this submission is relevant for this track for the reasons explained in point 1. Other papers with specific applications have been part of this track in the past [1]. For example CLIMODE, which explores the problem of physics-informed weather forecasting.
> [1] https://github.com/sherrylixuecheng/AI_for_Science_paper_collection/blob/main/iclr/iclr_2024.csv

---

> > ### Comment · Reviewer_oxLB · 2024-11-29
> >
> > The responses partly adress my concerns, I will keep the score given in the first round.

---

### Official Review · Reviewer_bAgN · 2024-11-03

**Soundness:** 3
**Presentation:** 3
**Contribution:** 3
**Rating:** 6
**Confidence:** 1

**Summary:**

The paper presents the Geometric Graph Track Finding (GGTF) method for particle track finding in collider experiments. GGTF combines data from various sub-detectors using a unified geometric algebra representation. The method employs the Geometric Algebra Transformer to efficiently handle diverse geometric input types, making it adaptable to different detector technologies. GGTF demonstrates high efficiency in reconstructing particle trajectories, particularly excelling with displaced and low-momentum tracks, surpassing traditional techniques.

**Strengths:**

The paper presents a highly original and impactful method for particle track finding, the Geometric Graph Track Finding (GGTF), which leverages geometric algebra and an equivariant graph neural network (GNN) to overcome limitations in traditional, detector-specific approaches. By unifying data from multiple sub-detectors through geometric algebra, GGTF enables a detector-agnostic solution adaptable to various complex detector configurations, including those proposed for FCC-ee. The thorough experimental validation on different detector concepts demonstrates GGTF’s high quality, showing clear improvements in tracking efficiency and accuracy for challenging cases like low-momentum particles and displaced tracks.

**Weaknesses:**

I have several concerns as follows:
1. What's the definition of CT! in L356？
2. It would be better if the authors could compare their approach with some other GNN methods.
3. It would improve the paper if the authors provided more detailed descriptions of the experimental data, such as dataset size and model training parameters.
4. Why did the authors choose HDBSCAN, and could they compare it with some other methods?

**Questions:**

I have listed my questions in weakness section.

---

> ### Author Response · Authors · 2024-11-20
>
> Thank you for the comments. We provide below answers to each of the questions. Overall, we believe this paper provides a new representation for ML approaches for tracking in particle physics, a new method, and extensive work on dataset generation and evaluation on different detectors which fits the track of “Application to physical sciences” of ICLR 2025.
>
> This work provides a generic approach which can cope with different track signatures in a single algorithm. In classical approaches, the different signatures (tracks originating from the interaction point, tracks with high or low momentum, etc) are usually handled by independent algorithms. Multiple algorithms require significant person-power for the development, validation and maintenance and necessitates dedicated prescriptions to combine them in the subsequent physics analyses. The method described here in this paper solves those issues.
>
> This work has shown that the GGTF method can cope with heterogeneous data from different subdetectors and can build tracks with all the available information at once. Classical approaches often rely on iterative processes where e.g. sub-tracks are first constructed using the hit from each subdetectors before being combined and re-fitted with the sub-tracks from other subsystems. Enabling the usage of all the information available at once is beneficial for performance. For instance, in the presence of beam induced background, using hits only from the vertex will create many fake tracks which will only be removed when combining with the outer tracker/drift chamber.
>
> This work targets future colliders, for which detector layouts are under construction. Classical methods usually have free parameters that have to be tuned to find optimal solutions for a given detector geometry and need to be re-tuned upon detector geometry changes. GGTF has a demonstrated fast development cycle (proven by the seamless application to two different detector concepts) and brings therefore the needed flexibility for detector development in view of future High Energy Physics experiments.
>
> Detailed answers to the questions:
> 1. CT conformal tracking (this was addressed in the text)
> 2. We consider different architectures to compare against GATr. We compare against Gravnet and Edge classification+a GNN based on interaction network layers, as proposed in  Lieret & DeZoort, 2024. Both GNNs  are trained with the same loss function as GATr and evaluated with the HDBSCAN clustering. The performance of Gravnet is compared to that of GATr and the heuristic baseline, the results are shown in the appendix. The performance is largely degraded as the training of the network does not result in good output embedding spaces. The results are even more extreme with the second approach (EC+interaction layers GNN), where the training does not converge and the clustering does not find relevant tracks. Therefore we can not produce a track efficiency for this architecture.  It is not clear why the performance in this case is degraded, as Lieret & DeZoort show good performance for high energy tracks, and we will continue to investigate this architecture.
> 3. The description of the dataset was provided in Appendix A.1. Specifics about the input format for the CLD detector are provided in A.1.1 (Input format) and for the IDEA detector in A.1.2. We also include a mention to the appendix in the main text in Sections 5.1 and 5.2 and give a summary of the dataset characteristics. The model training parameters are described in Appendix A.1.1 “Models”, A.1.1 “Training” for the CLD detector and   Appendix A.1.2 “Models”, A.1.2 “Training” for the IDEA detector.
> 4. We have performed a comparison of HDBSCAN, DBSCAN, and beta based object condensation clustering, we present the results in Figure 7. Originally, we chose HDBSCAN because it was able to capture clusters of different densities, unlike DBSCAN, as HDBSCAN performs DBSCAN over different epsilon parameters (a distance threshold to merge clusters), and integrates the result. Our result show that HDBSCAN has increased performance compared to DBSCAN and similar performance compared to the beta-based object condensation clustering. We choose to use the HDBSCAN algorithm as it is faster compared to the beta-based object condensation cluster.
>
> Additionally, we also included the following changes:
> - Improved model overview: we modified Figure 3 to include more details about the pipeline such as the extraction of the output embedding space from the multivectors, and a block that represents the construction of reconstructed tracks. The caption is also updated.
> - Improved description of the loss added to section 4.’Loss function’
> - Improved description of the final track reconstruction method in the new paragraph 4.’Final track reconstruction’
>
> -Three ablation comparisons of: the loss function, different clustering algorithms, different GNN architectures (added to section 4, appendix A.1.1 and appendix A.1.1 respectively)

---

> > ### Comment · Reviewer_bAgN · 2024-11-25
> >
> > Thanks for the response of author and I'll keep my rating.

---

### Official Review · Reviewer_fYKR · 2024-11-04

**Soundness:** 2
**Presentation:** 1
**Contribution:** 3
**Rating:** 5
**Confidence:** 4

**Summary:**

This paper works on a very unique application of machine learning, related to event trajectory reconstruction for particle physics experiments. Although work has been done on solving this problem in the past with machine learning, this work is detector agnostic. They heavily rely on a previously published graphical learning work, GATr, and modify it to suit their needs.

**Strengths:**

- The losses L_B, L_V that they introduced are novel and add integrate voltage prediction into the learning pipeline. The experimental results are also very convincing.

**Weaknesses:**

- This paper is not ready yet at its current iteration for a machine learning conference. The presentation isn’t very clear and I failed to understand the technically novel aspects of this paper. A decent portion of the paper (e.g. Figs. 1,2 ) doesn’t add anything to my understanding of the technically novel machine learning aspects of this paper.
- A lot of the time they refer the readers to read other papers to understand further important details of their paper. When writing papers, it is indeed very difficult to determine how much do the authors need to explain wrt the past works. However, in this situation, they could have used the space available to write more detailed explanations of their works. Please refer to the StyleGAN paper and some other papers using styleGAN to see how to present past SOTA work and using that to build your own case:
  - Analyzing and Improving the Image Quality of StyleGAN https://openaccess.thecvf.com/content_CVPR_2020/papers/Karras_Analyzing_and_Improving_the_Image_Quality_of_StyleGAN_CVPR_2020_paper.pdf
  - Encoding in Style: a StyleGAN Encoder for Image-to-Image Translation https://openaccess.thecvf.com/content/CVPR2021/papers/Richardson_Encoding_in_Style_A_StyleGAN_Encoder_for_Image-to-Image_Translation_CVPR_2021_paper.pdf

**Questions:**

- It will help to put this paper under a very granular machine learning category and write it from that perspective. I understand that this work is trying to do temporal graphical structure prediction, but it needs a few more details regarding what category it falls under: multi-modal learning? multi-Domain learning? Multi-stream learning? Data preprocessing and transformation? Cyclic time-series event detection? Etc.
- Please make it way more clear what losses are being used to train which aspects of their Graphical NN. Also, how the representations learned are being used to at run time to reconstruct the trajectories? Fig. 3 is supposed to do that but it is not helping at all.

---

> ### Author Response · Authors · 2024-11-20
>
> Thank you for your constructive suggestions.
> This paper was submitted to the track of ‘applications to physical sciences’ of ICLR, since the application is not as common as material property prediction [1], weather forecasting [2], or molecular design, we considered it was important to introduce the problem during the introduction (Fig 1,2) to a non-expert audience.  This approach has been used for other papers accepted in ICLR in the past under the same track, which also consider specific physics problems such as the ‘climate and weather forecasting’ [2].  Since this track is in the conference, we considered the contribution was appropriate as it represent a large leap forward for the particle-physics community by applying novel ML methods to solve a problem that typically has very long (multiple years) development cycles.
>
> We consider that the application of geometric (equivariant) graph neural network together with object condensation is novel as typically these architectures are used for node classification or graph regression/classification. The  representation of multiple sub detectors into a single format (the multivectors in the geometric algebra) is also a novelty of our work, that in other fields, such as molecule research, has been largely studied in ML conferences.
>
> Although there are ML papers dedicated to object detection, the usually focus on LiDAR datasets which have differences compared to our proposed dataset:
> - The number of hits is larger in the LiDAR based datasets, making architectures as GATr not applicable due to their lack of scalability
> - The geometry input types are only coordinates and not vectors or more complex geometries that appear in detectors
> - The particle physics dataset requires a large domain knowledge which is hard to bridge for a non-expert
> Overall, we believe the study of GNNs for object detection in particle-physics can push the understanding of ML architectures and techniques to new geometries and therefore is of interest to the ML community and in particular, to the physical sciences track of ICLR.
>
> We have improved the related work section, specially the ML-based tracking approaches.
>
> To address the questions:
> - Presenting the paper from the ML perspective would be an option, but we chose to develop the paper from the application side due to the track and since other papers from different applications have been previously presented in this conference, as can be seen in the list [3] , which contains 70 accepted papers in relation to physical sciences for ICLR 2024.
> - The only loss being used is the object condensation loss as described in section “4. Loss and final track reconstruction”. We create a new section that describes the loss exclusively and another section that discusses how the representation learned are used to reconstruct trajectories during inference. Figure 3 (now Figure 2) has also been updated  to include more details about the pipeline such as the extraction of the output embedding space from the multivectors, and a block that represents the construction of reconstructed tracks. The caption is also updated.
>
> Additionally, we also included three ablation comparisons of: the loss function, different clustering algorithms, different GNN architectures (added to section 4, appendix A.1.1 and appendix A.1.1 respectively)
>
>
> [1] Yan, Keqiang, et al. "Complete and efficient graph transformers for crystal material property prediction." arXiv preprint arXiv:2403.11857 (2024).
> [2] Verma, Yogesh, Markus Heinonen, and Vikas Garg. "Climode: Climate and weather forecasting with physics-informed neural odes." arXiv preprint arXiv:2404.10024 (2024).
> [3] https://github.com/sherrylixuecheng/AI_for_Science_paper_collection/blob/main/iclr/iclr_2024.csv

---

### Meta-Review · Area_Chair_oirW · 2024-12-17

**Metareview:**

This submission received two negative sores and a positive score after rebuttal. The reviewer thought that only part of her/his concerns was solved. After carefully reading the paper, the review comments, the AC can not recommend the acceptance of this submission, which is based the fact that the average score is under the threshold bar. The AC also recognizes the contributions confirmed by the reviewers, and encourages the authors to update the paper according to the discussion and submit it to a more relevant conference.

**Additional Comments On Reviewer Discussion:**

After discussion, while reviewer #oxLB thought that the responses did not fully address her/his concerns. Reviewer#oxLB did not responded to the rebuttal and kept the original negative score.

---

### Decision · Program_Chairs · 2025-01-22

Reject